# Glutamate oxaloacetate transaminase 1 is dispensable in macrophage differentiation and anti-pathogen response
Lishan Zhang ⓘ , Zhengyi Wu, Xuanhui Qiu, Jia Zhang & Shih-Chin Cheng ⓘ ✉

Macrophages play a pivotal role in orchestrating the immune response against pathogens. While the intricate interplay between macrophage activation and metabolism remains a subject of intense investigation, the role of glutamate oxaloacetate transaminase 1 (*Got1*) in this context has not been extensively assessed. Here, we investigate the impact of *Got1* on macrophage polarization and function, shedding light on its role in reactive oxygen species (ROS) production, pathogen defense, and immune paralysis. Using genetically modified mouse models, including both myeloid specific knockout and overexpression, we comprehensively demonstrate that *Got1* depletion leads to reduced ROS production in macrophages. Intriguingly, this impairment in ROS generation does not affect the resistance of *Got1* KO mice to pathogenic challenges. Furthermore, *Got1* is dispensable for M2 macrophage differentiation and does not influence the onset of LPS-induced immune paralysis. Our findings underscore the intricate facets of macrophage responses, suggesting that *Got1* is dispensable in discrete immunological processes.

The host innate immune response is paramount in defending against pathogens, with macrophages playing a central role in this process. Upon stimulation, macrophages undergo a series of activation states that dictates their functional fates. Classically activated (M1) pro-inflammatory macrophages exhibited a proinflammatory state and are typically induced by type I cytokine interferon-γ (IFNγ) and toll-like receptor (TLR) stimulation in the presence of microbial triggers. In contrast, alternative activated (M2) macrophages are induced by type II cytokines such as interleukin-4 (IL-4) and IL-13, promoting tissue remodeling and defending against parasite infection, especially helminths[1]. Distinct activation states are accompanied by differential metabolic phenotypes, both upon and following infection[2]. However, dysregulated macrophages activation can lead to compromised immune responses, resulting in prolonged infections and heightened susceptibility to pathogens. In conditions such as sepsis, an overwhelming cytokines storm can trigger septic shock and extensive tissue damage[3–5]. Conversely, overshooting activation of M1 macrophages could lead to immunosuppression and immune paralysis, rendering the immune system ineffective against subsequent microbial infections[4,6,7]. Furthermore, during the advanced stages of cancer, tumor-associated macrophages (TAMs) often adopt the M2 phenotype, promoting tumor growth, angiogenesis, and immunosuppression[8,9]. Macrophages also play a critical role in resolving inflammation, underscoring the need to unravel the intricate regulatory networks governing macrophage activation and function.

One of the crucial characteristics in M1 macrophages is the metabolic state shifts from oxidative phosphorylation (OXPHOS) to aerobic glycolysis[10–13]. M1 macrophages heavily consume glucose, generating substantial lactate while shifting mitochondria from ATP synthesis to ROS production[10,14]. This glycolytic shift is crucial for phagocytosis and production of inflammatory cytokines[15,16]. In contrast, M2 polarization relies on OXPHOS to ensure an adequate supply of ATP[17,18]. Concomitantly, fatty acid oxidation and glutamine metabolism pathways support the tricarboxylic acid (TCA) cycle in M2 macrophages[19–21].

Metabolic pathways, including glucose, fatty acid, and amino acid metabolism, converge at acetyl-CoA, a precursor for the TCA cycle. In M1 macrophages, impaired isocitrate dehydrogenase (IDH) and succinate dehydrogenase (SDH) functions lead to increased succinate levels due to the aspartate-argininosuccinate shunt (AAS), which connects the TCA cycle with the urea cycle[16]. The urea cycle is pivotal for macrophage NO production, while the AAS facilitates the connection between TCA cycle intermediates and the urea cycle. Within the AAS, glutamate oxaloacetate

State Key Laboratory of Cellular Stress Biology, School of Life Science, Faculty of Medicine and Life Sciences, Xiamen University, Xiamen 361102, China.
✉e-mail: jamescheng@xmu.edu.cn

aminotransferase (GOT) utilizes pyridoxal phosphate as a cofactor to catalyze the mutual transformation between aspartate and oxaloacetate, as well as between α-ketoglutarate and glutamate[16].

GOT exists in two isoforms, GOT1 and GOT2. Aminooxy acetic acid (AOAA), a GOT inhibitor, disrupts the malate-aspartate shuttle, reducing mitochondrial glycolytic substrate utilization[22]. AOAA preconditioning shifts M1 macrophages towards greater reliance on mitochondrial respiration, resembling M0 macrophages in terms of aerobic glycolysis[16]. Notably, AOAA inhibits M1 polarization, leading to reduced nitric oxide (NO) and inducible nitric oxide synthase (iNOS) levels, as well as suppressed IL-6 expression[16]. Cytokine and NO production are crucial for pathogen clearance during infections[23,24]. These findings suggest that AOAA-induced disruption of the aspartate-malate shuttle could perturb the balance between the Krebs cycle and NO cycle, potentially impacting to succinate dehydrogenase (SDH) function and overall cellular processes.

Here, our study shows that AOAA pre-treatment suppresses the differentiation of macrophages into the M1 phenotype, aligns with previous research[16]. However, our investigation reveals that genetic knockout of Got1 in macrophages does not exert any influence on the differentiation of pro-inflammatory macrophages. Meanwhile, we conducted experiments involving mice with myeloid cell-specific overexpression of Got1, and the results indicate that Got1 is not essential for macrophage inflammatory responses. Furthermore, whether Got1 is overexpressed or knocked out in myeloid cells, it has no impact on the differentiation of M2 macrophages induced by IL-4. Within LPS-induced tolerance assay, Got1 also exhibits a passive role, devoid of active involvement, as evident from the results. These findings present a contrasting perspective, challenging the previously assumed role of Got1 in the macrophage inflammatory response.

## Results

### AOAA-induced cytokines production inhibition is independent of *Got1*

AOAA, a known inhibitor of GOT1, has been identified for its ability to curtail cytokines production in macrophages[16]. Here, we examine whether AOAA-induced cytokine reduction is mechanistically linked to GOT1 inhibition. To assess this, we generated $Got1^{\Delta LysM}$ mice by breeding $Got1$-floxed mice with $Lyz2$-Cre mice, resulting in myeloid cell-specific $Got1$ depletion (Supplementary Fig. 1a, b). Genotype of mice was validated by genomic DNA PCR (Supplementary Fig. 1c). Confirmation of $Got1$ depletion efficiency was achieved through evaluating cellular protein and messenger RNA (mRNA) levels in bone marrow-derived macrophages (BMDMs) (Supplementary Fig. 1d, e). Subsequently, we explored whether AOAA treatment in BMDMs differentiated from $Got1^{\Delta LysM}$ mice could reduce cytokines production (Fig. 1a). Remarkably, AOAA effectively suppressed IL-6 but not TNFα protein expression levels not only in BMDMs from $Got1^{f/f}$ mice but also in BMDMs from $Got1^{\Delta LysM}$ mice (Fig. 1b, c). In addition, NO is generated by iNOS within pro-inflammatory macrophages, playing a crucial role as a notable marker of pro-inflammatory reactions[25]. AOAA dose-dependently reduced nitric oxide (NO) and inducible nitric oxide synthase (iNOS) levels[16], in alignment in our data indicating a reduction in NO levels in both BMDMs from $Got1^{f/f}$ and $Got1^{\Delta LysM}$ mice (Fig. 1d). As there are two isoforms of $Got$, we are intrigued by the possibility of $Got2$ expression being upregulated in response to AOAA, potentially serving as a compensatory mechanism against $Got1$'s inhibitory effects. However, the mRNA and protein levels of $Got1$ and $Got2$ remained unchanged upon AOAA treatment (Supplementary Fig. 2a–c). PF-04859989, another inhibitor reported to specifically targeted GOT1[26], was employed for comparison in our experiments. Our results indicate that PF-04859989, like AOAA, demonstrates comparable inhibitory effects on IL-6

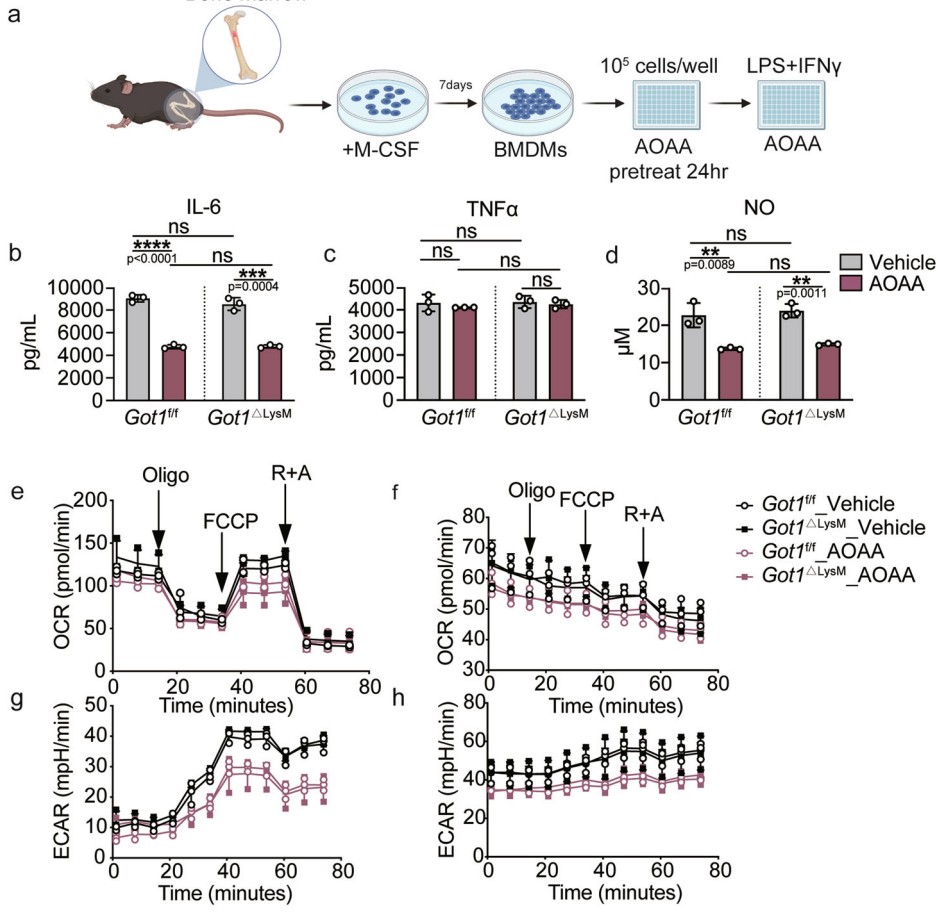

**Fig. 1 | The inhibition of AOAA-induced cytokine production and metabolic states in BMDMs is independent of *Got1*. a** Schematic of AOAA treatment in BMDMs differentiated from $Got1^{f/f}$ and $Got1^{\Delta LysM}$ mice. **b**, **c** Supernatant IL-6, TNFα protein levels of BMDMs derived from $Got1^{f/f}$ and $Got1^{\Delta LysM}$ mice. BMDMs were pre-treated with 10 mM AOAA and then stimulated with 10 mM AOAA and LPS (20 ng/mL) + IFNγ (100 ng/mL) for 24 h. $Got1^{f/f}$ mice, n = 3 biologically independent experiments, $Got1^{\Delta LysM}$ mice, n = 3 biologically independent experiments. **d** NO production level in BMDMs derived from $Got1^{f/f}$ and $Got1^{\Delta LysM}$ mice. Seahorse analysis of OCR in M0 macrophages (**e**) and M1 macrophages (**f**), ECAR in M0 macrophages (**g**) and M1 macrophages (**h**). ns, p > 0.05, not significant, **p < 0.01, ***p < 0.001, ****p < 0.0001, unpaired, p values were determined by two-tailed Student's t test. Data are representative of three independent experiments (mean ± SD).

**Fig. 2 | *Got1* deficiency fails to influence the differentiation of pro-inflammatory macrophages.**
**a** Schematic of pro-inflammatory macrophages differentiation. **b–d** *Il6*, *Tnfa* and *Il1b* mRNA levels in BMDMs stimulated with LPS (100 ng/mL) for 4 h or LPS (20 ng/mL) + IFNγ (100 ng/mL) for 24 h. *Got1*^f/f mice, *n* = 3 biologically independent experiments; *Got1*^ΔLysM mice, *n* = 3 biologically independent experiments. **e, f** IL-6, TNFα protein levels in the supernatant determined by ELISA. **g** NO production level in BMDMs stimulated with LPS (100 ng/mL) or LPS (20 ng/mL) + IFNγ (100 ng/mL) for 24 h. *Got1*^f/f mice, *n* = 3 biologically independent experiments; *Got1*^ΔLysM mice, *n* = 3 biologically independent experiments. **h** Lactate production levels in BMDMs stimulated with LPS (100 ng/mL) or LPS (20 ng/mL) + IFNγ (100 ng/mL) for 24 h. *Got1*^f/f mice, *n* = 3 biologically independent experiments; *Got1*^ΔLysM mice, *n* = 3 biologically independent experiments. ns, *p* > 0.05, not significant, unpaired, two-tailed Student's *t* test. Data are representative of three independent experiments (mean ± SD).

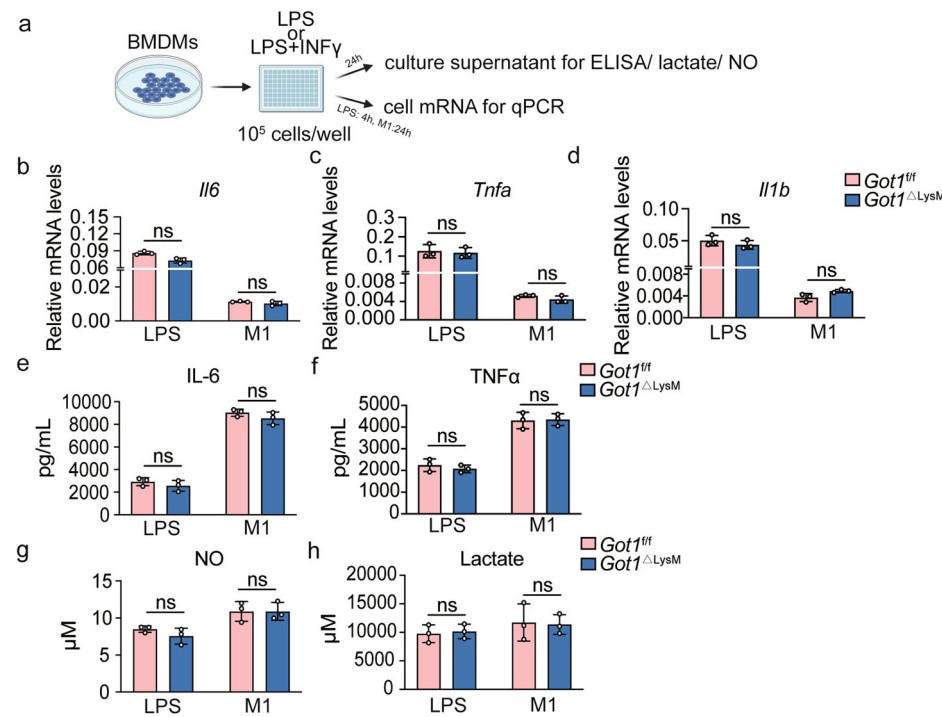

and NO production, while having no impact on TNFα production (Supplementary Fig. 2d–f). Moreover, knockdown of *Got2* by siRNA (sequences show in Supplementary Table 1) does not impact cytokine production upon LPS stimulation (Supplementary Fig. 3). Concurrently, we evaluated oxygen consumption rate (OCR) and extracellular acidification rate (ECAR) levels in BMDMs derived from both *Got1*^f/f and *Got1*^ΔLysM mice. Additionally, our finding reveals that AOAA (Fig. 1e–h) or PF-04859989 (Supplementary Fig. 4a–h), not only suppresses OCR and ECAR in BMDMs from *Got1*^f/f mice but also exerts similar effects in BMDMs from *Got1*^ΔLysM mice. These results collectively underscore the pleiotropic effects of AOAA or PF-04859989, extending beyond the inhibition of GOT1 enzymatic activity. Their impact resonates significantly in modulating the production of inflammatory cytokines and NO not only in BMDMs from *Got1*^f/f mice but also those from *Got1*^ΔLysM mice.

## The differentiation of pro-inflammatory macrophages is independent of *Got1*

To investigate the role of *Got1* in pro-inflammatory macrophages, we therefore stimulated BMDMs with LPS or a combination of LPS and IFNγ to induce pro-inflammatory macrophages (Fig. 2a). Both LPS-stimulated and LPS + IFNγ-stimulated macrophages exhibited anticipated increases in inflammatory cytokines with significant elevation in mRNA (*Il6, Tnfa, Il1b*) and protein (IL-6, TNF-α) expression levels[27]. Surprisingly, *Got1* deficiency in BMDMs from *Got1*^ΔLysM mice did not influence the mRNA (*Il6, Tnfa, Il1b*) and protein (IL-6, TNF-α) production, when compared to *Got1*^f/f mice (Fig. 2b–f). Furthermore, M1 macrophages primarily utilize glycolysis as their main metabolic pathway, generating significant levels of lactate[28,29]. However, the NO and lactate production remained unaffected by *Got1* depletion (Fig. 2g, h).

To validate our findings, we isolated peritoneal macrophages (PMs), recognized as LyzM+ cells within the peritoneal cavity[30,31], from both *Got1*^f/f or *Got1*^ΔLysM mice and repeated the same experiments as described in BMDMs (Supplementary Fig. 5a). We intraperitoneally injected 3% thioglycolate per mouse to stimulate the generation of an ample population of peritoneal macrophages (PMs) in mice, and subsequently harvested the PMs after a period of 3-4 days. Consistent with BMDMs, the mRNA (*Il6, Tnfa, Il1b*) and protein (IL-6, TNF-α) levels upon stimulation with LPS or LPS + IFNγ were comparable in both *Got1*^f/f or *Got1*^ΔLysM mice

(Supplementary Fig. 5b–i). Since LPS is a TLR4 agonist, we investigated whether stimulation with an intact pathogen using a non-TLR4 agonist would yield different results. To address this, PMs were stimulated with Heat-killed fungal pathogen *C. albicans* (HKCA) while there was still no difference between *Got1*^f/f or *Got1*^ΔLysM mice (Supplementary Fig. 5b–d).

The AAS is pivotal in M1 macrophages[10,32]. GOT1, an AAS enzyme, catalyzes the interconversion between aspartate and oxaloacetate, as well as α-ketoglutarate and glutamate (Fig. 3a)[33,34]. The knockout of GOT1 resulted in a significant accumulation of aspartate. However, this effect was not observed in the AOAA-treated group[35]. Aspartate metabolism is crucial in M1 macrophages, as evidenced by their depletion of aspartate and its metabolic products[32]. Nonetheless, it remains unclear whether aspartate metabolism is associated with macrophage differentiation. Interestingly, we observed that pre-treatment with aspartate resulted in a significant increase in *Il1b* mRNA levels when inducing pro-inflammatory macrophages (Fig. 3b, c). Given the importance of aspartate metabolism, we hypothesized the accumulation of *Got1* could potentially modify the functionality of M1 macrophages. We then employed the distinctive attributes of the *Rosa26* locus to facilitate the construction of transgenic mouse model with *Got1*[36]. As shown in Supplementary Fig. 6a, b, we crossbreed *Got1*^stop/+ mice with *Lyz2-Cre* mice to obtain mice with myeloid cell-specific overexpression of *Got1*. Genotyping was performed by extracting genomic DNA from mouse tail tissue (Supplementary Fig. 6c). Then GFP reporter was detected by flow cytometry (Supplementary Fig. 6d). Cellular mRNA was extracted in BMDMs to assess the expression level *Got1* (Supplementary Fig. 6e). BMDMs derived from *Got1*^stop/+ or *Got1*^stop/+; *Lyz2-Cre* mice were then stimulated with HKCA, LPS or a combination of LPS and IFNγ to induce pro-inflammatory macrophages. The mRNA (*Il6, Tnfa, Il1b*) and protein (IL-6, TNF-α) levels remained unaffected by *Got1* overexpression in response to various stimulations (Fig. 3d–k).

Taken together, our comprehensive approach utilizing *Got1* conditional knockout and *Got1* overexpression models demonstrates unequivocally that *Got1* is dispensable for the differentiation of pro-inflammatory macrophages.

### *Got1* depletion decreases ROS production in macrophages without affecting pathogen defense

We next explored the potential impact of *Got1* on reactive oxygen species (ROS) production in macrophages, a critical component of the

**Fig. 3 | Got1 overexpression fails to influence the differentiation of pro-inflammatory macrophages. a** Model of GOT1 in malate-aspartate shuttle. **b** Schematic of Aspartate treatment in wildtype BMDMs stimulated with LPS (100 ng/mL) or LPS (20 ng/mL) + IFNγ (100 ng/mL). **c** *Il1b* mRNA levels in BMDMs stimulated with LPS (100 ng/mL) for 4 h or LPS (20 ng/mL) + IFNγ (100 ng/mL) for 24 h. *n* = 3 biologically independent experiments. **d–f** *Il6*, *Tnfa* and *Il1b* mRNA levels in BMDMs stimulated with LPS (20 ng/mL) + IFNγ (100 ng/mL) for 24 h. *Got1*^stop/+ mice, *n* = 3 biologically independent experiments; *Got1*^stop/+; *Lyz2-Cre* mice, *n* = 3 biologically independent experiments. **g–i** *Il6*, *Tnfa* and *Il1b* mRNA levels in BMDMs stimulated with LPS (100 ng/mL) or HKCA (HKCA: BMDM = 1:1) for 4 h. *Got1*^stop/+ mice, *n* = 3 biologically independent experiments; *Got1*^stop/+; *Lyz2-Cre* mice, *n* = 3 biologically independent experiments. **j, k** IL-6, TNFα protein levels in the supernatant determined by ELISA. BMDMs were stimulated with 100 ng/mL LPS, LPS (20 ng/mL) + IFNγ (100 ng/mL) or HKCA (HKCA: BMDM = 1:1) for 24 h. *Got1*^f/f mice, *n* = 3 biologically independent experiments; *Got1*^ΔLysM mice, *n* = 3 biologically independent experiments. ns, *p* > 0.05, not significant, *\*p* < 0.05, *\*\*p* < 0.01, unpaired, two-tailed Student's *t* test. Data are representative of three independent experiments (mean ± SD).

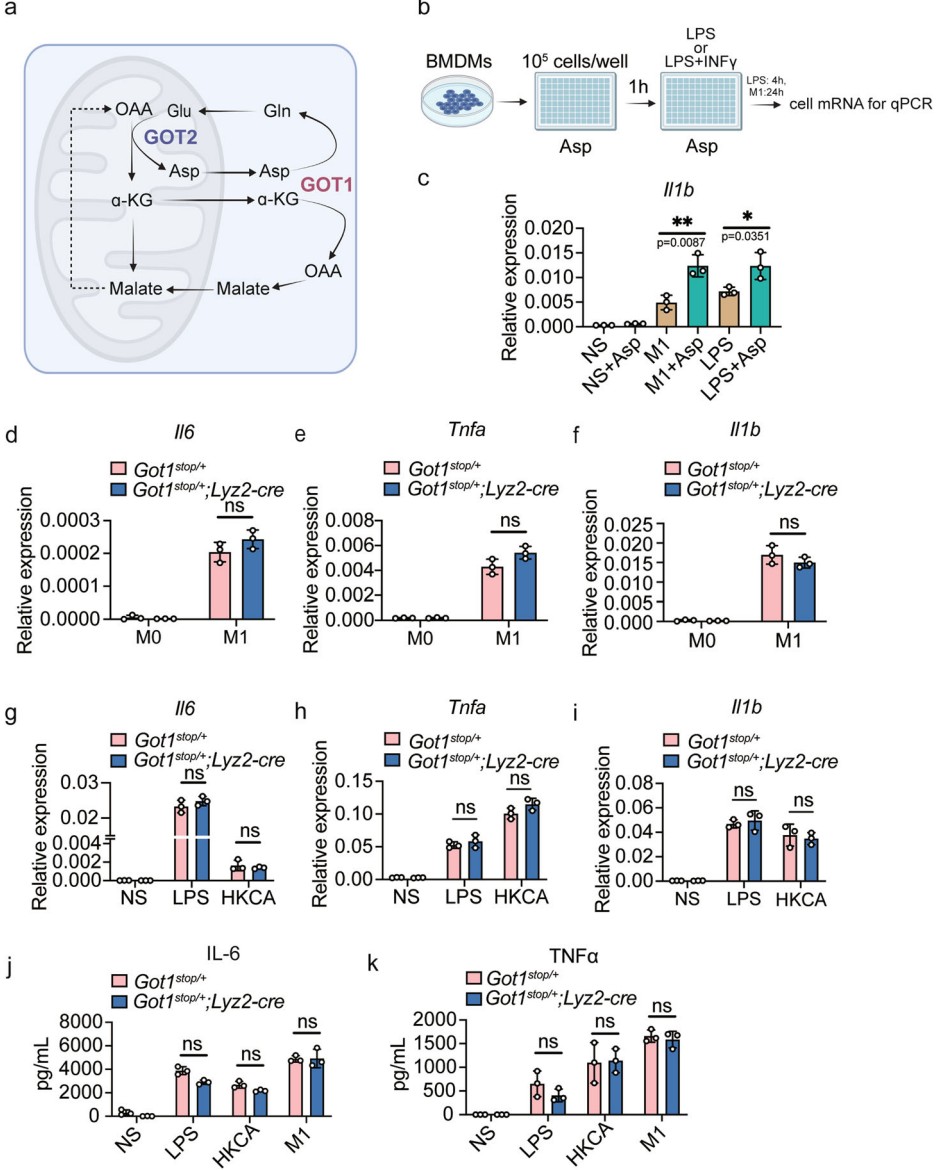

innate immune response against invading pathogens[37,38]. Our experimental setup (Fig. 4a) included the use of 2′,7′-Dichlorodihydrofluorescein diacetate (H2DCFDA) as an indicator of ROS generation, with the ROS inhibitor N-acetyl-l-cysteine (NAC) as a negative control[39,40]. BMDMs derived from *Got1*^ΔLysM mice exhibited lower DCF-DA fluorescence levels indicating compromised ROS production capacity due to *Got1* depletion (Fig. 4b). Nevertheless, no alterations in ROS production were detected in BMDMs derived from *Got1*^stop/+ or *Got1*^stop/+; *Lyz2-Cre* mice (Supplementary Fig. 6f, g). Intriguingly, this reduction in ROS levels did not affect the phagocytosis (Fig. 4c) or bactericidal abilities of BMDMs against *Staphylococcus aureus* (*S. aureus*) and *Escherichia coli* (*E. coli*) (Fig. 4d). Furthermore, we pretreated BMDMs with either AOAA (Fig. 4b) or PF-04859989 (Supplementary Fig. 7). Interestingly, our results showed that regardless of the presence of inhibitors, the deletion of *Got1* effectively suppressed ROS production. Similarly, PMs from *Got1*^ΔLysM mice displayed lower DCF-DA fluorescence levels without any alteration in phagocytosis activity (Supplementary Fig. 5j, k). Next, we performed pathogens infection in vivo to test the role of *Got1*-mediated ROS production in *Got1*^f/f and *Got1*^ΔLysM mice (Fig. 4e). It is foreseeable that there is no difference in survival between *Got1*^f/f and *Got1*^ΔLysM mice following infections with *S. aureus* or *Candida albicans* (*C. albicans*) (Fig. 4f, g). Together, our

findings reveal that *Got1* depletion decreases macrophage ROS production without impacting the ability to combat pathogens effectively.

## The differentiation of M2 macrophages is independent of Got1

M2 macrophages rely on an intact tricarboxylic acid (TCA) cycle and are distinct from their pro-inflammatory counterparts[13,15,16,41]. α-ketoglutarate (αKG) plays a crucial role in the alternative (M2) activation of macrophages[21]. Previous metabolomic studies have indicated that the deficiency of GOT1 results in a significant reduction in intracellular αKG levels (Xu et al.). Motivated by these findings, we investigated the role of Got1 in IL-4-induced M2 macrophage differentiation. We assessed mRNA samples from BMDMs derived from *Got1*^f/f and *Got1*^ΔLysM mice treated with IL-4 for 24 h as the schematic shown in Fig. 5a. The mRNA expression levels of M2 markers, including *Chil3*, *Arg1*, and *Retnla* were evaluated[42,43]. Remarkably, *Got1* deficiency in BMDMs did not influence M2 macrophage differentiation, as evidenced by the unchanged mRNA expression of these marker (Fig. 5b–d). Consistently, PMs from *Got1*^ΔLysM mice exhibited no differences in M2 marker expression upon IL-4 stimulation (Fig. 5e–g). Therefore, we pretreated BMDMs or PMs derived from *Got1*^f/f and *Got1*^ΔLysM mice with AOAA (Fig. 5b–g) or PF-04859989 (Supplementary Fig. 8a–f), followed by IL-4 induction to drive BMDM or PMs towards M2 differentiation. Interestingly, our results demonstrate that both AOAA and PF-

**Fig. 4 | *Got1* deficiency decreases ROS production in BMDMs without affecting pathogen defense.**
**a** Schematic of ROS detection in macrophages. **b** ROS production in BMDMs. Prior to ROS detection, BMDMs were pre-treated with AOAA for 24 h, then treated with HKCA (HKCA: BMDM = 1:1) for 1 h. *Got1*^f/f mice, *n* = 3 biologically independent experiments; *Got1*^ΔLysM mice, *n* = 3 biologically independent experiments. **c** Phagocytosis of BMDMs. FITC labeled-HKCA was utilized for phagocytosis assay with BMDMs. **d** CFU Count of *S. aureus* and *E. coli* in BMDMs-In-Vitro-Killing Assay. **e** Schematic of infection design. **f, g** Percentage of mice survival. ns, *p* > 0.05, not significant, **p* < 0.05, ***p* < 0.01, unpaired, *p* values were determined by two-tailed Student's *t* test in **b**–**d** and log-rank test was used for survival comparison in **f** and **g**. Data are representative of three independent experiments (mean ± SD).

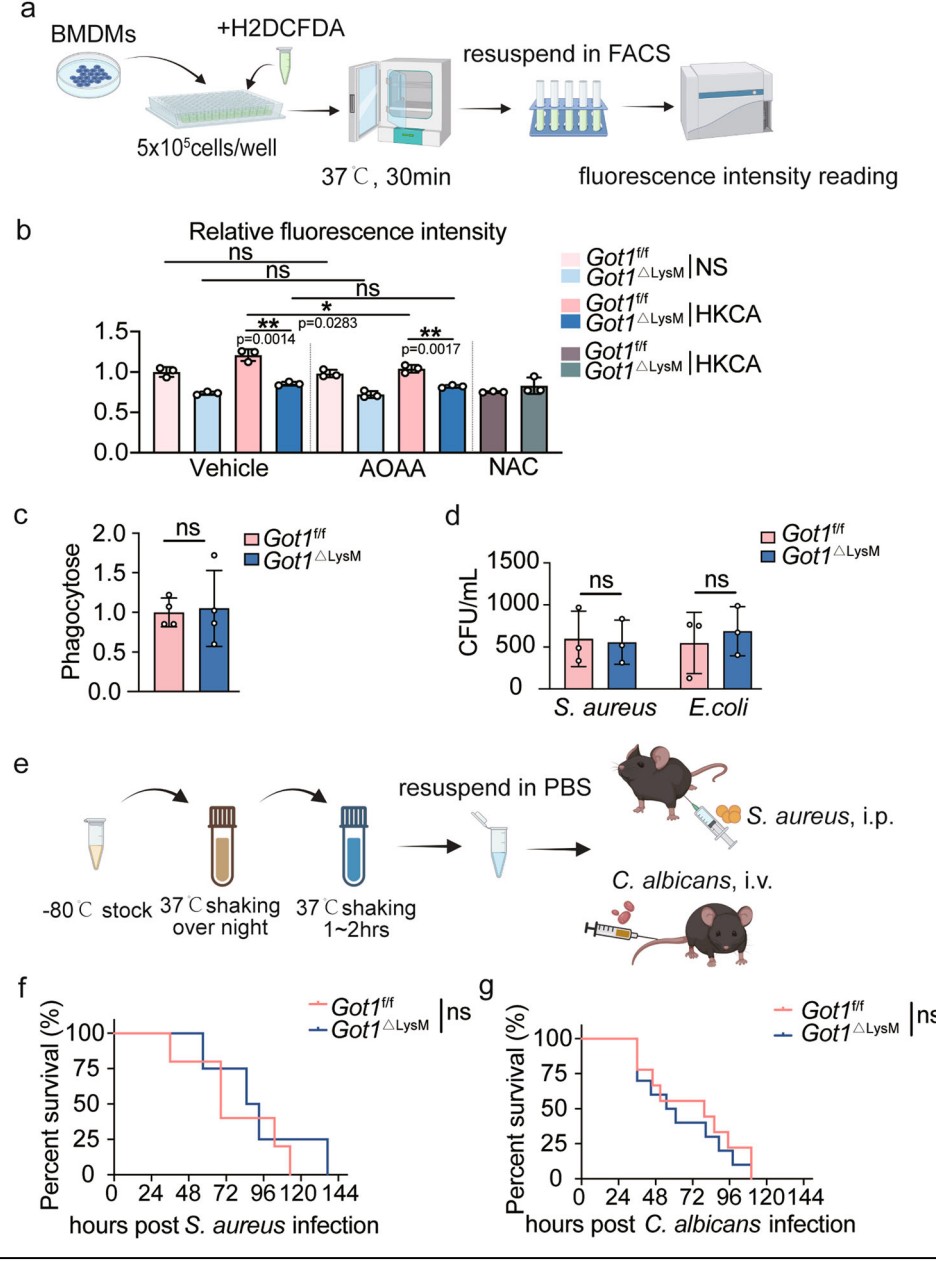

04859989 exert inhibitory effects on M2 differentiation in BMDMs or PMs, although AOAA did not exhibit inhibition of *Chil3* and *Arg1* in BMDMs (Fig. 5b, c). Moreover, overexpression of *Got1* in macrophages, as observed in *Got1*^stop/+ or *Got1*^stop/+; *Lyz2-Cre* mice, did not affect the mRNA expression levels of M2 markers (Fig. 5h–j). These data collectively establish that the differentiation of M2 macrophages is independent of *Got1*.

## LPS induced immune tolerance in macrophages is independent of *Got1*

Sepsis-induced immunosuppression carries risks for adverse outcomes[7,44]. Metabolic reprogramming towards aerobic glycolysis, known as the Warburg effect, is observed in human peripheral blood monocytes following LPS stimulation, often associated with mitochondrial damage[45,46]. To understand whether *Got1* is involved in LPS-induced immune tolerance, we employed an in vitro model using BMDMs derived from *Got1*^ΔLysM or *Got1*^stop/+; *Lyz2-Cre* mice (Fig. 6a). Remarkably, neither *Got1* knockout nor overexpression in BMDMs alters the protein (IL-6, TNF-α) levels (Fig. 6b, c, f, g), lactate levels (Fig. 6d) and NO levels (Fig. 6e) of LPS-induced immune tolerance. Furthermore, we pretreated BMDMs with AOAA (Fig. 6b–e) or PF-04859989 (Supplementary Fig. 9a–d). The results reveal that both

AOAA and PF-04859989 exhibit inhibitory effects on the production of IL-6 by BMDMs upon the single LPS stimulation, with no impact on TNFα production. Additionally, we unexpectedly observed that pretreatment with AOAA slightly enhances the production of IL-6 and TNFα upon the LPS retreatment (Fig. 6b, c). These results firmly establish that *Got1* does not play a role in the inflammatory response underlying LPS-induced immune paralysis in macrophages.

## Discussion

The intricate interplay between macrophage activation and metabolic reprogramming constitutes a pivotal axis governing their dynamic engagement in immune responses. Within this intricate network, the role of *Got1* is less explored and warrants further investigation. Our study endeavors to discern the multifaceted implications of *Got1* in macrophage polarization and function.

The perturbations within the TCA cycle within M1 macrophages, characterized by dual disruptions at the IDH and SDH nodes, underpin citrate and itaconic acid accumulation, concomitant with heightened succinate levels. This sequential perturbation amplifies the aspartate-argininosuccinate shunt, consequently invigorating the urea cycle and

**Fig. 5 | The differentiation of M2 macrophages is independent of *Got1*. a** Schematic of the differentiation of M2 macrophages. **b–d** *Chil3*, *Arg1*, and *Retnla* mRNA levels in BMDMs, derived from *Got1^f/f* and *Got1^ΔLysM* mice, pretreated with 10 mM AOAA for 24 h, then treated with IL-4 (20 ng/mL) for 24 h. *n* = 3 biologically independent experiments. **e–g** *Chil3*, *Arg1*, and *Retnla* mRNA levels in PMs, derived from *Got1^f/f* and *Got1^ΔLysM* mice, pretreated with 10 mM AOAA for 24 h, then treated with IL-4 (20 ng/mL) for 24 h. *n* = 3 biologically independent experiments. **h–j** *Chil3*, *Arg1*, and *Retnla* mRNA levels in BMDMs, derived from *Got1^stop/+* and *Got1^stop/+*; *Lyz2-Cre* mice, treated with IL-4 (20 ng/mL) for 24 h. *n* = 3 biologically independent experiments. ns, *p* > 0.05, not significant, *\**p* < 0.05, *\*\**p* < 0.01, *\*\*\**p* < 0.001, unpaired, *p* values were determined by two-tailed Student's *t* test. Data are representative of three independent experiments (mean ± SD).

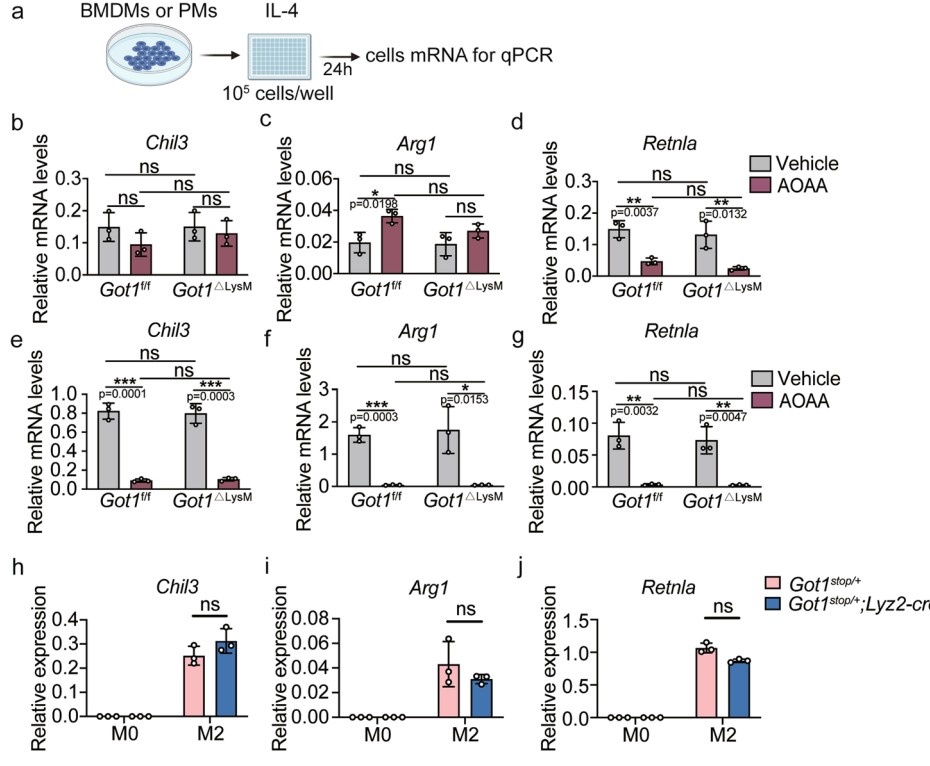

**Fig. 6 | LPS induced immune tolerance in macrophages is independent of *Got1*. a** Schematic of immune tolerance induced with LPS. **b, c** IL-6, TNFα protein levels in the supernatant of *Got1^f/f* and *Got1^ΔLysM* mice derived BMDMs were determined by ELISA. Before LPS retreatment, BMDMs were pretreated with 10 mM AOAA for 24 h, *n* = 3 biologically independent experiments. **d** Lactate production level in BMDMs stimulated with LPS (100 ng/mL) retreatment for 24 h. **e** NO production level in BMDMs stimulated with LPS (100 ng/mL) retreatment for 24 h. **f, g** IL-6, TNFα protein levels in the supernatant of *Got1^stop/+* and *Got1^stop/+*; *Lyz2-Cre* mice derived BMDMs were determined by ELISA in. *n* = 3 biologically independent experiments. ns, *p* > 0.05, not significant, *\**p* < 0.05, *\*\**p* < 0.01, *\*\*\**p* < 0.001, unpaired, *p* values were determined by two-tailed Student's *t* test. Data are representative of three independent experiments (mean ± SD).

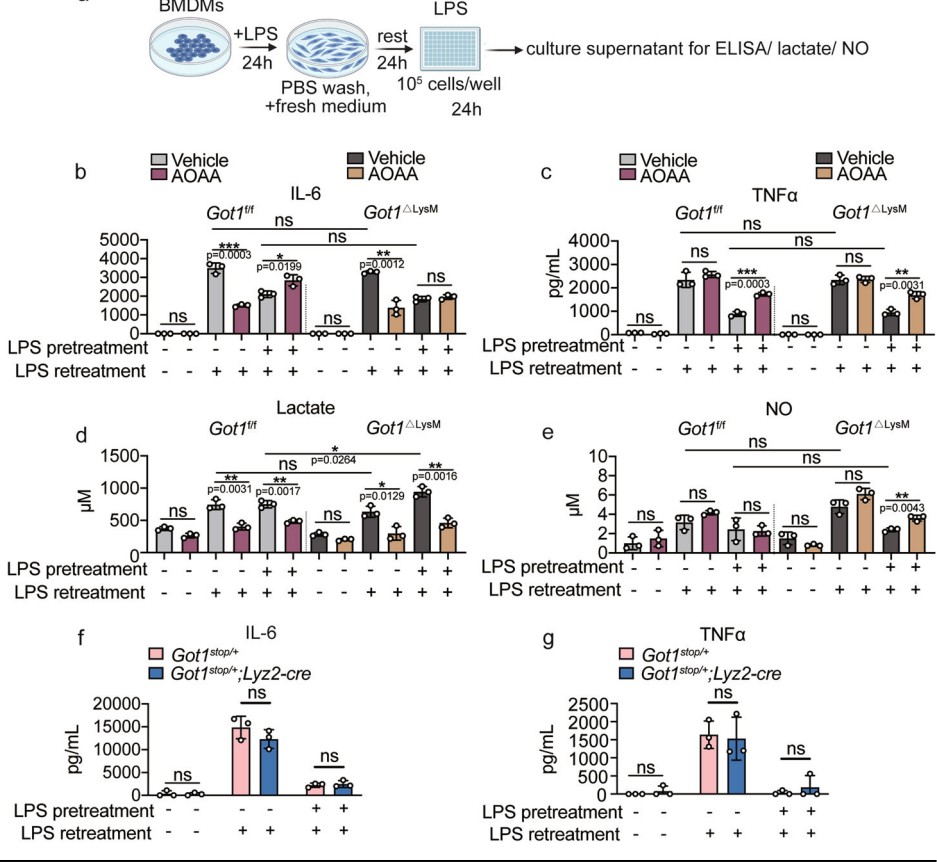

facilitating NO production[10,16,47]. Of note, Abhishek K. Jha *et al.* demonstrated that treating macrophages with AOAA can inhibit the pro-inflammatory phenotype of M1 macrophages by suppressing GOT1[16], leading to reduced IL-6 and NO production. Correspondingly, our findings substantiate AOAA-mediated suppression of M1 differentiation, culminating in concordant diminution of IL-6 protein levels

and NO release within BMDMs following LPS stimulation. This concurrence underscores the regulatory role of the AASS pathway in the inflammatory response, albeit in another way (Fig. 1b–d).

AOAA has gained widespread recognition as a selective GOT1 inhibitor in various contexts[16,34,48–51]. Nonetheless, our investigation transcends the conventional confines of AOAA's influence, shedding light on its potential multifaceted effects that extend beyond GOT1 inhibition. Notably, the study by Xu et al. in 2017 unveils the pivotal role of *Got1* in contributing to the dynamic equilibrium between Th17 and Treg subsets[49]. Given that *Got1* is critical in the conversion of glutamate to α-ketoglutarate (α-KG), AOAA's modulation of GOT1 activity attains paramount significance. Crucially, inhibition of GOT1 by AOAA counters the suppressive impact exerted by 2-hydroxyglutarate (2-HG) on *Foxp3* expression, potentially facilitating Treg differentiation[49]. Conversely, the further investigation by Xu et al. broadens the implications of AOAA, transcending its traditional role as a specific GOT1 inhibitor[35]. Their meticulous exploration, utilizing *Got1*$^{f/f}$; *CD4-Cre* mice, introduces intriguing complexities by revealing divergent outcomes in T cells exposed to AOAA treatment. This intriguing finding propels an in-depth exploration into AOAA's comprehensive mechanisms of action, raising pertinent questions regarding the scope of its specificity as a GOT1 inhibitor. Our own data, utilizing both AOAA and PF-04859989 to inhibit GOT1 activity in both *Got1*$^{f/f}$ and *Got1*$^{ΔLysM}$ BMDMs, reveal that both inhibitors effectively inhibit cytokine production regardless of *Got1* expression. This suggests a pleiotropic effect of current GOT1 inhibitors. Therefore, further research is warranted to develop more specific GOT1 inhibitors for studying GOT1 effects using inhibitors accurately.

Crucially, the viability of BMDMs remained unaltered upon AOAA treatment[16]. This intervention induced significant reduction in cytokine and NO production within M1 macrophages, a trend consistent across our observations in our *Got1*$^{f/f}$ and *Got1*$^{ΔLysM}$ mice (Fig. 1b–d). This intriguing outcome hints at the potential participation of the AASS pathway in this modulation, albeit leaning more towards an outcome rather than a causal factor. The precise cellular target of AOAA's influence in macrophages prompts intricate considerations. Our study primarily observed a decrease in the generation of reactive oxygen species (ROS) in macrophages, yet this phenomenon appeared inconsequential to the overall antimicrobial defense exhibited by the mice. Such duality underscores an intricate compensatory interplay between *Got1* and *Got2*. Notably, we gauged the mRNA expression of *Got2* to ascertain any compensatory response in the absence of *Got1* in macrophages, revealing no overt alterations (Supplementary Fig. 10). Moreover, we knocked down *Got2* in BMDMs using siRNA. The experimental results were not surprising: knocking down *Got2* did not affect the production of inflammatory factors in LPS-stimulated macrophages (Supplementary Fig. 3). Furthermore, our observations underscore the limited involvement of *Got1* in IL-4-induced M2 macrophages (Fig. 5) and its role in LPS-induced immune tolerance (Fig. 6).

In summary, our investigations collectively unravel that the anti-inflammatory effects elicited by AOAA in macrophages manifest independently of *Got1*. Consequently, the intrinsic implication of *Got1* in shaping the broader panorama of macrophage inflammatory responses appears to be of a lesser magnitude.

## Methods
### Generation of animals
The female or male mice used in our study were all C57BL/6J strain, aged between 6 and 12 weeks. These mice were housed in a controlled environment of the Laboratory Animal Center at Xiamen University, a facility that adheres to SPF-level standards and maintains a regulated day-night light cycle. *Got1*$^{f/f}$ and Lyz2-Cre mice were procured from Gempharmatech Co., Ltd, whereas *Got1*$^{stop/+}$ mice were generated by Xiamen University Animal Center. To ensure experimental consistency, all mice included in our study were carefully selected to be sex and age-matched, originating from the same litters. All mouse experiments were conducted in accordance with the guidelines and regulations set forth by the Xiamen University Laboratory Animal Center and

were approved by the Institutional Animal Care and Use Committee. We have complied with all relevant ethical regulations for animal use.

### Generation and manipulation of BMDMs
For BMDMs differentiation, bone marrow cells were obtained from the femur and tibia of 6-12-week-old C57BL/6J mice and cultured in complete DMEM medium containing 10% FBS and 40 ng/μL M-CSF (Novoprotein, CD34). On day 3, half of the culture volume was replenished with complete medium containing 40 ng/μL M-CSF, and the cells were cultured until day 7 to yield BMDMs.

For LPS-induced pro-inflammatory macrophages polarization, $10^5$ BMDMs per well were seeded in 96-well-plate, and then stimulated with 100 ng/mL LPS (Lipopolysaccharides from E. coli, Invivogen # tlrl-pb5lps). After 4 h of LPS stimulation, cells were lysed for mRNA extraction. Culture supernatants were collected 24 h post-LPS stimulation to assess cytokine production by ELISA.

For M1 macrophage polarization, $10^5$ BMDMs per well were seeded in a 96-well-plate and treated with 20 ng/mL LPS + 100 ng/mL IFNγ (Novoprotein, CM41). Cell lysis and culture supernatants were collected 24 h after LPS + IFNγ stimulation.

For M2 macrophage polarization, $10^5$ BMDMs per well were seeded in a 96-well plate and treated with 20 ng/mL IL-4 (Novoprotein, CK74). Cell lysis was collected 24 h after IL-4 treatment.

Regarding HKCAs treatment, $10^5$ BMDMs per well were seeded in a 96-well-plate, and HKCAs were added at a comparable quantity to the BMDM cells. Cell lysis was collected 4 h after HKCAs stimulation for mRNA extraction, and culture supernatants were collected 24 h after HKCAs stimulation for ELISA analysis.

### Isolation and cultivation of peritoneal macrophages
To obtain peritoneal macrophages, we intraperitoneally injected 3 mL 3% thioglycolate per mouse to stimulate the generation of an ample population of PMs in mice. After a period of 3-4 days, we injected 4 mL of PBS containing 1% FBS into the mouse peritoneal cavity for harvesting the PMs. After gently massaging the abdomen and incubating on a shaker for 5 min, we collected peritoneal fluid using a syringe, repeating the process twice. The fluid was then centrifuged at 1500 g for 5 min at 4 °C, and the supernatant was discarded. If necessary, red blood cells were lysed with ACK buffer. For ACK lysis (1.5 M NH$_4$Cl, 100 mM KHCO$_3$, 10 mM EDTA-2Na), 3 mL of ACK buffer was added to each sample and incubated for 3 minutes at room temperature, followed by termination with PBS in volumes greater than twice that of ACK. After centrifugation, peritoneal cells were plated in cell culture dishes. Following a 6–8-h incubation, non-adherent cells were removed by PBS washing, leaving adherent peritoneal macrophages. These macrophages were subsequently detached, counted, and seeded in plates for subsequent experiments.

### Real-time RT-PCR
The mRNA extraction from cells was performed as describe in reference[20]. BMDMs were treated as the requirement, followed by treatment with 200 μL of RNA lysis buffer. Magnetic beads conjugated with oligo-dT18 were added for mRNA extraction. Subsequently, the extracted mRNA underwent reverse transcription using a mixture of dNTPs (Beyotime, D7366), oligo-dT, RNA transcriptase (Accurate Biology, AG11605) and RNase inhibitor (Accurate Biology, AG11608) to yield cDNA. The obtained cDNA was employed for real-time PCR analysis with 2× SYBR (Accurate Biology, AG11701). The relative mRNA expression levels were assessed utilizing the $2^{-ΔCT}$ method with Ct values, employing *B2m* as the endogenous reference gene. The primer sequences are available in Supplementary Table 2.

### Western blot
The cultured medium of BMDMs was removed and cells were washed with PBS. BMDMs were then lysed by RIPA buffer containing a protease inhibitor cocktail. After that, 5× SDS-loading buffer was added and cell lysate samples were boiled at 95 °C for 5 min. Subsequently, 10 μL of the sample

was carefully loaded into the wells of an SDS-PAGE gel for protein separation according to their molecular weights. The separated proteins were then transferred onto a Polyvinylidene fluoride (PVDF) membrane. The PVDF membrane was meticulously blocked by immersion in a solution of 5% non-fat milk for 1 h at room temperature. Following blocking, the PVDF membrane was incubated overnight at 4°C with the primary antibody against GOT1 (1:1000 dilution, Abclonal A11363), GOT2 (1:1000 dilution, Abclonal A19245) and β-Actin (1:1000 dilution, Abcam ab8226). The subsequent day, the primary antibody was carefully removed, and the PVDF membrane was gently washed three times with PBST (Phosphate Buffered Saline with 1% Tween-20). The PVDF membrane was then subjected to incubation at room temperature for 1 h with the secondary antibody (1:5000 dilution, SAB L3012), followed by another three washes with PBST. For protein visualization, HyperSignal ECL kit (4A Biotech Co. Ltd, 4AW012-500) was used and Chemiluminescent Imaging and Analysis System (ChampChemi™ Professional+) was used for images analyzation.

## ELISA

$10^5$ cells per well were seeded in a 96-well-plate. Pro-inflammatory macrophage differentiation was induced by adding either 100 ng/mL LPS or $10^5$ HKCA for 24 h. For M1 macrophage polarization, a combination of 20 ng/mL LPS and 100 ng/mL IFNγ was applied for 24 h. After this incubation, culture supernatants were collected from each well for cytokine level assessment. IL-6 (Invitrogen; 88-7064-88) and TNFα (Invitrogen; 88-7324-88) levels were quantified following the respective ELISA kit instructions.

## Lactate and NO

For lactate level detection, the collection of cell culture supernatant followed the same procedure as the preparation before conducting the ELISA assay. Subsequently, the cell culture supernatant was diluted with PBS and mixed with a mixture containing lactate oxidase (Sigma L0638), Amplex Red (Alfachem 119171-73-2), and HRP in a 1:1 ratio. The mixture was incubated for approximately 10–20 min at room temperature. Fluorescence values were then measured at an excitation wavelength of 528 nm and an emission wavelength of 590 nm.

For NO level detection, the collection of cell culture supernatant followed the same procedure as the preparation before conducting the ELISA assay. In the cell culture, NO production results in the formation of nitrite ions through reactions with oxygen and water. These nitrite ions further react with sulfanilamide to create a diazonium salt, which can react with N-1-naphthylethylenediamine dihydrochloride. We prepared a mixture by combining Solution A (ethylenediamine dihydrochloride) and Solution B (sulfanilamide) in a 1:1 ratio. Subsequently, we mixed this mixture with the cell culture medium in a 1:1 ratio to facilitate the reaction. The level of nitrite ions serves as an indicator of NO production. Finally, we measured the absorbance of the reaction solution at 540 nm.

## ROS

Prior to utilization, BMDMs were washed with either PBS or FBS-free DMEM twice. For $Got1^{\Delta LysM}$ mice, $5 \times 10^5$ BMDMs were seeded per well in 96-well-U-shaped-bottom-plate and 5 μM DCF-DA working solution was added in each well, followed by incubation at 37°C in the dark for 30 min. Subsequently, the cells were washed twice with PBS, resuspended by FACS buffer. The Fluorescence intensity was measured by CytoFLEX Flow Cytometer. For $Got1^{stop/+}$; Lyz2-Cre mice, $2.5 \times 10^4$ BMDMs were seeded per well in 96-well cell culture plate. After 1.5 h, each well received an amount of HKCA double the number of cells. Phagocytosis occurred for 30 min, following which HKCA was discarded. Subsequently, 200 uL of reaction solution, comprising 5 uM Amplex Red and 0.1 U/mL HRP, was added to each well. The reaction was carried out at room temperature in a light-avoidant environment, and fluorescence measurements were taken.

## Pathogen in vitro killing assay and in vivo infection

*S. aureus*, *E. coli* and *C. albicans* were cultivated to an optical density (OD600) of approximately 0.8 following overnight incubation at 37°C, 220 rpm for

12 h. For *S. aureus* and *E. coli*, we calibrated an OD600 reading of 0.5 to correspond to $2.5 \times 10^8$ CFU. For *C. albicans*, CFU were determined using a serial dilution method, followed by manual counting under a microscope.

For in vitro killing assay, $2 \times 10^5$ BMDMs per well were seeded in 24-well-plate. Next, *S. aureus* was added into BMDMs at multiplicity of infection (MOI) of 1, while *E. coli* was added into BMDMs at MOI of 10. The cultures were carried out without antibiotics. After a 30-min incubation for phagocytosis, extracellular bacteria were removed by washing with PBS containing 400 ng/mL gentamicin. Then, we supplemented them with DMEM complete medium containing 400 ng/mL gentamicin and allowed the BMDMs to continue incubating for an additional 2 h. After this period, we washed the cells with PBS, lysed them with sterile water, and released the intracellular bacteria. The cell lysates were then stepwise diluted using PBS and subsequently plated and cultured onto LB agar plates at 37°C overnight.

In our in-vivo infection experiments, *S. aureus* and *C. albicans* were cultivated as above. *S. aureus* infection was administered intraperitoneally (*i.p.*) with 5000 CFU per mouse. *C. albicans* infection was administered intravenously (*i.v.*) with $2 \times 10^5$ CFU per mouse. The survival of the mice was meticulously monitored over a 7-day period.

## Statistics and reproducibility

All statistical samples were carried out in accordance with the descriptions provided in the main text. All data were analyzed with the two-tailed Student's $t$ test method by GraphPad Prism software. All quantified data are represented as mean ± standard deviation (SD). "*" indicates $P$-value less than 0.05 and "**" indicates $P$-value less than 0.01, "***" indicates $P$-value less than 0.001, "****" indicates $P$-value less than 0.0001, while "ns" indicates $P$-value more than 0.05 and have no significant difference.

## Data availability

The data supporting the graphs presented in the paper are available in the Supplementary Data section. All siRNA sequences are available in Supplementary Table 1. All qPCR primer sequences are available in Supplementary Table 2. The gating strategy of all flow cytometry plots were shown in Supplementary Fig. 11. All of the unedited images in western blotting were shown in Supplementary Fig. 12. All of the unedited DNA agarose gel images in genotyping were shown in Supplementary Fig. 13.

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

## Acknowledgements

This study was supported by the following fundings: National Natural Science Foundation of China grants 32161133020 and 32070904 (S.-C.C.), Fundamental Research Funds for the Central Universities 20720220003 (S.-C.C.), start-up fund of Xiamen University (S.-C.C.), Natural Science Foundation of Xiamen province of China 3502 Z20227003 (Zhang Jia). All schematic diagrams and illustrations in this manuscript were created using BioRender (BioRender.com).

## Author contributions

S.-C.C. conceived the study, L.Z. and S.-C.C. designed the experiments and analyzed the data. L.Z., X.Q., Z.W. performed the experiments. J.Z. assisted in designing the experiments and analyzing the data. The research received funding acquisition and support by S.-C.C. All authors discussed the results, L.Z. wrote the original draft and S.-C.C. reviewed and edited the manuscript.

## Competing interests

The authors declare no competing interests.
