## [Peer Review File · Communications Biology]

Reviewers' comments:

Reviewer #1 (Remarks to the Author):

This is a sound and well-conducted piece of work that comprehensively demonstrates that the transaminase GOT1 is dispensable in several models of macrophage activation. While the fact that GOT1 is dispensable is the focus of this paper, perhaps the more interesting aspect of this work is the discrepancy between the use of AOAA as a pharmacological inhibitor of GOT1 (and other transaminases) and the use of genetic models targeting GOT1, related to some of the recent work performed by the group of Jonathan Powell in T cells. Therefore, I feel that more work should be conducted to further clarify the differences in these approaches. My specific comments are as follows:

1. In line 82, the authors describe AOAA as a 'known inhibitor of GOT1'. AOAA is a pan-transaminase inhibitor, and specific GOT1 inhibitors do exist (PMID 31787239). Could the authors perhaps include a specific GOT1 inhibitor alongside AOAA in figure 1 in order to further emphasise that AOAA is not a suitable compound to target GOT1 specifically?
2. The authors should also further consider the effect of GOT2 inhibition. Inhibition of GOT2 in LPS-stimulated macrophages would be expected to exert modulation of cytokine release, potentially through fumarate accumulation (PMID 36890227). Some mention of this as a possible explanation for AOAA's effects would be appropriate
3. Figure 1h-j should be moved to the supplementary data
4. As mentioned before, more work should be done to directly compare the use of AOAA to genetic manipulation of GOT1. Can the authors perform some metabolic assays to this end, perhaps measuring ECAR and OCR in *Got1^{f/f}* and *Got1^{lysm}* macrophages, and then compare this to AOAA-treated macrophages and macrophages treated with a specific GOT1 inhibitor. Inhibition of the AAS in the Jha et al paper (PMID 25786174) reduced the LPS-induced increase in ECAR, and it would be very informative to investigate whether this still occurs in *Got1* knockout macrophages. This would also complement similar metabolic profiling performed in a study in T cells (PMID 36726001), more discussion of which is warranted in this manuscript
5. Similarly, can the authors also include AOAA (and perhaps a specific GOT1 inhibitor) in the assays performed to measure ROS production (Fig. 4b), alternative macrophage activation (Fig. 5b-g) and trained immunity (Fig. 6b-e)
6. In figure 1, data from *Got1^{f/f}* macrophages and *Got1^{lysm}* macrophages should be depicted on the same graph. For example, data from Fig 1b and 1c should be on the same graph and the same for Fig. 1e and 1f
7. In general, the data throughout the study are quite convincing. However, the data in Fig. 2b-g are quite variable and have large error bars which may be masking any potential differences. This also

indicates that there may be some technical issues with the stimulations. The authors might consider repeating some of these experiments.

8. In Fig. 4b, it appears that some of the conditions (HKCA + NAC) only have two replicates, yet a statistical test has been performed on these two conditions. Could the authors please clarify?

9. The rationale for the experiments performed in Fig. 3 is that aspartate accumulates upon inhibition of GOT1. However, the authors do not demonstrate that this is indeed the case, which would be important to show as rationale for performing these experiments in the first place.

10. It is also slightly unclear as to what is the rationale for examining the role of GOT1 in M2 polarisation, given that the aspartate-argininosuccinate shunt was originally shown to be important specifically in M1 macrophages. Again, can the authors perform metabolic assays in *Got1^{f/f}* and *Got1^{lysm}* macrophages in the context of IL-4 stimulation?

11. Can Fig S4f be formatted in the same way as Fig. 4b for the sake of consistency? Showing the data as histograms depicting relative fluorescence intensity is clearer

Reviewer #2 (Remarks to the Author):

This study explores the role of glutamate oxaloacetate transaminase 1 (*Got1*) in the immune response, particularly in macrophage activation and metabolism. *Got1* is a pyridoxal phosphate-dependent enzyme with important roles in amino acid metabolism. Using genetically modified mouse models, the investigators find that the depletion of *Got1* in macrophages reduces reactive oxygen species (ROS) production but surprisingly does not affect resistance to either *S. aureus* or *C. albicans*. Similarly, the authors found no role for *Got1* in cytokine production or macrophage polarization in response to LPS and IL-4. Notably, these findings are contradictory to some of the observations reported by the Artyomov group (Jha et al., 2015).

While the experimental approach, utilizing both pharmacological and genetic tools, was methodologically sound, the overall results are less captivating due to the absence of any observations indicating a tangible impact of *Got1* on immune responses studied. As such, the functional relevance of *Got1* remains ambiguous.

Response to reviewer

Reviewer #1 (Remarks to the Author):

This is a sound and well-conducted piece of work that comprehensively demonstrates that the transaminase GOT1 is dispensable in several models of macrophage activation. While the fact that GOT1 is dispensable is the focus of this paper, perhaps the more interesting aspect of this work is the discrepancy between the use of AOAA as a pharmacological inhibitor of GOT1 (and other transaminases) and the use of genetic models targeting GOT1, related to some of the recent work performed by the group of Jonathan Powell in T cells. Therefore, I feel that more work should be conducted to further clarify the differences in these approaches. My specific comments are as follows:

In line 82, the authors describe AOAA as a 'known inhibitor of GOT1'. AOAA is a pan-transaminase inhibitor, and specific GOT1 inhibitors do exist (PMID 31787239). Could the authors perhaps include a specific GOT1 inhibitor alongside AOAA in figure 1 in order to further emphasise that AOAA is not a suitable compound to target GOT1 specifically?

Response:

We appreciate the suggestion provided by the reviewer. We conducted experiments utilizing the GOT1 inhibitor (PF-04859989) referenced in the article (PMID 31787239). Our investigations, involving the induction of BMDMs into M1 phenotype using LPS + $\text{INF}\gamma$, revealed notable inhibitory effects of PF-04859989 on NO and IL-6 production in both *Got1*^{fllox} and *Got1* ^{Δ LysM} mice, while TNF- α levels remained unaffected (Response Figure 1).

Furthermore, as elucidated in the by Jonathan Powell's group (PMID 36726001), it's worth noting that AOAA does not directly affect *Got1* but rather modulates T cell differentiation through chemical reactions involving keto acids. Consequently, we posit that inhibitors such as AOAA and PF-04859989 exhibit pleiotropic effects beyond the inhibition of the target GOT1, as discussed in PMID 36726001. We have integrated these new findings into Fig. S2d-f and addressed them in the revised manuscript (lines 102-111).

Response figure 1. F-04859989 showed inhibitory effects on NO and IL6 in *Got1*^{fllox} and *Got1* ^{Δ LysM} mice, while TNF- α remained unaffected.

2. The authors should also further consider the effect of GOT2 inhibition. Inhibition of GOT2 in LPS-stimulated macrophages would be expected to exert modulation of cytokine release, potentially through fumarate accumulation (PMID 36890227). Some mention of this as a possible explanation for AOAA's effects would be appropriate

Response: We appreciated the suggestion provided by the reviewer and have undertaken additional experiments to shed light on the role of *Got2* in the differentiation process of LPS-stimulated macrophages. We knocked down *Got2* in BMDMs by siRNA and induced LPS-stimulated macrophage differentiation. Our findings reveal that the knockdown of *Got2* does not exert any discernible impact on the expression levels of IL-6 and TNF α in LPS-stimulated macrophages (Response Figure 2). To reflect these new findings, we have integrated the data into the

revised version of Fig. S3 and have elaborated on them within the manuscript (lines 95-96).

Response figure 2 *Got2* does not impact the expression of IL-6 and TNF α in LPS-stimulated macrophages, both at the protein and mRNA levels.

3. Figure 1h-j should be moved to the supplementary data.

Response: We have moved these figures to the supplementary data in Fig. S2a-c.

4. As mentioned before, more work should be done to directly compare the use of AOAA to genetic manipulation of GOT1. Can the authors perform some metabolic assays to this end, perhaps measuring ECAR and OCR in *Got1^{f/f}* and *Got1^{ΔLysM}* macrophages, and then compare this to AOAA-treated macrophages and macrophages treated with a specific GOT1 inhibitor. Inhibition of the AAS in the Jha et al paper (PMID 25786174) reduced the LPS-induced increase in ECAR, and it would be very informative to investigate whether this still occurs in *Got1* knockout macrophages. This would also complement similar metabolic profiling performed in a study in T cells (PMID 36726001), more discussion of which is warranted in this manuscript

Response: We highly appreciate the valuable suggestion provided by the reviewer. We conducted experiments wherein *Got1^{f/f}* and *Got1^{ΔLysM}* BMDMs were subjected to LPS+INF γ treatment, preceded by pretreatment with both AOAA and PF-04859989. Intriguingly, our results demonstrate that both AOAA and PF-04859989 effectively inhibited OCR and ECAR not only in *Got1^{f/f}* BMDMs but also in *Got1^{ΔLysM}* BMDMs (Response Figure 3). This observation underscores the pleiotropic effects of current *Got1* inhibitors, extending beyond the inhibition of *Got1* activity, as previously reported by Jonathan Powell in T cells. These data have been integrated into Fig. 1 and Fig. S4. Additionally, to enhance clarity, we have included a thorough discussion on AOAA's effects in T cells in lines 235-250 of the manuscript.

Response figure 3 AOA and PF-04859989 inhibited OCR and ECAR values in both *Got1*^{fl/fl} and *Got1*^{ΔLysM} BMDMs.

5. Similarly, can the authors also include AOAA (and perhaps a specific GOT1 inhibitor) in the assays performed to measure ROS production (Fig. 4b), alternative macrophage activation (Fig. 5b-g) and trained immunity (Fig. 6b-e)

Response: In response to the reviewer's valuable suggestion, we have conducted additional experiments outlined below:

1. *Got1^{flox}* and *Got1^{ΔLysM}* BMDMs were pre-treated with AOAA or PF-04859989, followed by stimulation with heat-killed *Candida albicans* (HKCA) to assess ROS production. Our findings indicate that irrespective of inhibitor treatment, the knockout of *Got1* led to a reduction in ROS production (Response Figure 4). This highlights that AOAA and PF-04859989 do not exclusively target *Got1* to exert their effects. The newly acquired data have been integrated into the current version of Fig. 4b and Fig. S7 (line 169-171).

Response figure 4 *Got1* depletion decreases ROS production in BMDMs.

2. *Got1^{flox}* and *Got1^{ΔLysM}* BMDMs were pre-treated with AOAA or PF-04859989, followed by IL-4 treatment to induce M2 polarization in BMDMs or peritoneal macrophages (PMs). Our results reveal that both AOAA (Response Figure 5) and PF-04859989 (Response Figure 6) exert inhibitory effects on M2 differentiation, although AOAA did not inhibit *Arg1* expression. The newly obtained data have been incorporated into the current version of Fig. 5b-g and Fig. S8 (line 192-196).

Response figure 5 AOAA exerts inhibitory effects on BMDMs M2 differentiation, although did not exhibit inhibition of *Arg1*.

Response figure 6 PF-04859989 exerts inhibitory effects on BMDMs M2 differentiation.

3. In our *in vitro* model of LPS-induced tolerance experiments, BMDMs were pre-treated with AOAA or PF-04859989 before LPS restimulation. Our results demonstrate that both AOAA and PF-04859989 exhibit inhibitory effects on IL-6 production by BMDMs upon the single LPS stimulation, with no impact on TNF production. Additionally, we unexpectedly observed that pretreatment with AOAA slightly enhances the production of IL-6 and TNF upon LPS retreatment (Response Figure 7 and 8). The newly acquired data have been integrated into the current version of Fig. 6b-e and Fig. S9 (line 209-213).

Response figure 7 The inhibitory effects of AOAA on the BMDMs upon LPS induced tolerance.

Response figure 8 The inhibitory effects of PF-04859989 on the BMDMs upon LPS induced tolerance.

These additional experiments offer further insights into the effects of AOAA and PF-04859989 on various cellular processes and have been comprehensively incorporated into the revised manuscript.

6. In figure 1, data from *Got1^{fl/fl}* macrophages and *Got1^{ΔLysM}* macrophages should be depicted on the same graph. For example, data from Fig 1b and 1c should be on the same graph and the same for Fig. 1e and 1f

Response: We have integrated the data from both *Got1^{fl/fl}* and *Got1^{ΔLysM}* into a unified graph in Fig. 1b-c.

7. In general, the data throughout the study are quite convincing. However, the data in Fig. 2b-g are quite variable and have large error bars which may be masking any potential differences. This also indicates that there may be some technical issues with the stimulations. The authors might consider repeating some of these experiments.

Response: We have addressed this concern by adding additional technical replicates, combining the data, and conducting a meticulous reanalysis. The outcomes of this reevaluation are depicted in Response Figure 9.

Response figure 9 *Got1* deficiency fails to influence the differentiation of pro-inflammatory macrophages.

8. In Fig. 4b, it appears that some of the conditions (HKCA + NAC) only have two replicates, yet a statistical test has been performed on these two conditions. Could the authors please clarify?

Response: The reviewer's observation regarding Fig. 4b is duly noted, and we apologize for any confusion arising from the statistical analysis oversight. Following this feedback, we thoroughly reviewed our data and conducted additional experiments to address the issue. Moreover, we have expanded the scope of our investigation by including the treatment of AOAA and PF-04859989 inhibitors. We pre-treated *Got1*^{flox} and *Got1*^{ALysM} BMDMs with AOAA or PF-04859989, followed by stimulation with heat-killed *Candida albicans* (HKCA) to assess ROS production. Our findings reveal that irrespective of inhibitor treatment, the knockout of *Got1* led to a reduction in ROS production (Please refer to Response Figure 4 above). This further supports the notion that AOAA and PF-04859989 do not exclusively target GOT1 to exert their effects. The newly obtained data have been integrated into the current version of Fig. 4b and Fig. S7 for clarity and completeness.

9. The rationale for the experiments performed in Fig. 3 is that aspartate accumulates upon inhibition of GOT1. However, the authors do not demonstrate that this is indeed the case, which would be important to show as rationale for performing these experiments in the first place.

Response: We appreciated the reviewer's insightful suggestion. In line with the rationale for our experiments in Fig. 3, we referenced the findings to Xu et al. (PMID 36726001), where metabolomic analysis was conducted in Wild type and *Got1* knockout. From these results, it was observed that the absence of *Got1* indeed led to the accumulation of aspartate (Response Figure 10). Additionally, the authors treated T cells with AOAA and found that AOAA treatment did not result in aspartate accumulation. This further strengthens the argument that AOAA does not exert its effects solely through the inhibition of GOT1. We have incorporated this clarification in lines 136-138 of the current version of the manuscript.

Response figure 10 Metabolomic data of WT, WT+AOAA, and GOT1^{-/-} in the article referenced by PMID 36726001.

10. It is also slightly unclear as to what is the rationale for examining the role of GOT1 in M2 polarisation, given that the aspartate-argininosuccinate shunt was originally shown to be important specifically in M1 macrophages. Again, can the authors perform metabolic assays in *Got1*^{f/f} and *Got1*^{lys} macrophages in the context of IL-4 stimulation?

Response: We appreciate the reviewer's insightful suggestion. In the study referenced by PMID 36726001, the authors conducted metabolomic analysis with and without GOT1 knockout. Remarkably, this analysis unveiled a significant decrease in α -ketoglutarate levels in T cells under both GOT1 knockout and AOA treatment conditions. This intriguing finding piqued our interest in investigating the relationship between GOT1 and M2 macrophages, particularly considering previous reports highlighting α -ketoglutarate's role in inducing M2 differentiation (PMID: 28714978). We have addressed this rationale and provided clarification in lines 182-185.

Response figure 10 Metabolomic data of WT, WT+AOAA, and GOT1^{-/-} in the article referenced by PMID 36726001.

11. Can Fig S4f be formatted in the same way as Fig. 4b for the sake of consistency? Showing the data as histograms depicting relative fluorescence intensity is clearer

Response: We reformatted Fig. S4f in a manner consistent with Fig. 4b, using histograms to depict relative fluorescence intensity for improved clarity (Response Figure 11b). The updated data have been integrated into the current version of Fig. S6g for enhanced consistency and coherence.

Response figure 11 Overexpression Got1 failed to influence the ROS production.

Reviewer #2 (Remarks to the Author):

This study explores the role of glutamate oxaloacetate transaminase 1 (Got1) in the immune response, particularly in macrophage activation and metabolism. Got1 is a pyridoxal phosphate-dependent enzyme with important roles in amino acid metabolism. Using genetically modified mouse models, the investigators find that the depletion of Got1 in macrophages reduces reactive oxygen species (ROS) production but surprisingly does not affect resistance to either *S. aureus* or *C. albicans*. Similarly, the authors found no role for Got1 in cytokine production or macrophage polarization in response to LPS and IL-4. Notably, these findings are contradictory to some of the observations reported by the Artyomov group (Jha et al., 2015).

While the experimental approach, utilizing both pharmacological and genetic tools, was methodologically sound, the overall results are less captivating due to the absence of any observations indicating a tangible impact of Got1 on immune responses studied. As such, the functional relevance of Got1 remains ambiguous.

Response: We appreciate the reviewer's thoughtful evaluation of our manuscript. We acknowledge that these results might appear less captivating, especially considering some contradictory observations reported by the Artyomov group (Jha et al., 2015). However, we believe our study sheds light on the potential limitations of using inhibitors, which may have pleiotropic effects and could lead to misleading conclusions. Additionally, the use of knockout mice could complement these findings, providing a more comprehensive understanding of *Got1*'s genuine function in physiology and pathophysiology.

Overall, while the functional relevance of *Got1* in the immune response remains ambiguous based on our current findings, we hope our study prompts further exploration and critical evaluation of *Got1*'s role in immune regulation.

Reviewers' comments:

Reviewer #1 (Remarks to the Author):

I commend the authors for the work that they have carried out in response to my previous comments. The manuscript is now much improved with these additions. There are however a few formatting/data analysis issues which should be addressed prior to publication:

1. Some of the graphs which depict both AOAA-treated cells and genetic models are missing important statistical comparisons which must be performed in order to test the claims made in the text. For example, in Figures 1b, c, 5b-g, 6, S4, statistics must be performed comparing the Got1f/f data to Got1lysm data. At the moment, the only tests shown are the comparisons between vehicle and AOAA. The reverse is true of Fig. 4b which requires the addition of statistical comparisons between vehicle and AOAA treatment
2. In line 130, there is a reference to Fig S5 which I think should be to Fig S4
3. If I am not mistaken, Fig. S4 is not actually referenced in the text?

Response to reviewer

I commend the authors for the work that they have carried out in response to my previous comments. The manuscript is now much improved with these additions. There are however a few formatting/data analysis issues which should be addressed prior to publication:

1. Some of the graphs which depict both AOAA-treated cells and genetic models are missing important statistical comparisons which must be performed in order to test the claims made in the text. For example, in Figures 1b, c, 5b-g, 6, S4, statistics must be performed comparing the *Got1^{f/f}* data to *Got1^{ΔLysM}* data. At the moment, the only tests shown are the comparisons between vehicle and AOAA. The reverse is true of Fig. 4b which requires the addition of statistical comparisons between vehicle and AOAA treatment

Response: We greatly appreciate your feedback and the opportunity to enhance the rigor of our manuscript. In response to your observations, we have conducted comprehensive statistical analyses comparing the *Got1^{f/f}* data to *Got1^{ΔLysM}* data across all relevant figures. Moreover, we have included additional statistical comparisons between vehicle and AOAA treatments in Figure 4b to fully address your concern.

During this process, we discovered an error in Figure 6e where the data for Vehicle and AOAA treatments were swapped. We have corrected this mistake and ensured the accuracy of all data presentations in the revised manuscript.

2. In line 130, there is a reference to Fig S5 which I think should be to Fig S4

Response: Thank you for your attention to detail. We have corrected this referenced error in the current new version. We appreciate your vigilance and have ensured the accuracy of these references in the revised manuscript.

3. If I am not mistaken, Fig. S4 is not actually referenced in the text?

Response: Thank you for pointing this out. Figure S4 was indeed referenced previously in line 105 of our manuscript in the previous version. We have highlighted and clarified this reference in line 107-109 of the current revision to ensure it is easily identifiable.

REVIEWERS' COMMENTS:

Reviewer #1 (Remarks to the Author):

The authors have answered all of my queries, I have no further comments.

Reviewer #1 (Remarks to the Author):

The authors have answered all of my queries, I have no further comments.

We thank you for all the efforts in helping us to improve our manuscript.